# Empower Nested Boolean Logic via Self-Supervised Curriculum Learning

**Hongqiu Wu**[1,2] and **Linfeng Liu**[1,2] and **Hai Zhao**[1,2*] and **Min Zhang**[3,4]

[1]Department of Computer Science and Engineering, Shanghai Jiao Tong University
[2]Key Laboratory of Shanghai Education Commission for Intelligent Interaction
and Cognitive Engineering, Shanghai Jiao Tong University, Shanghai, China
[3]School of Computer Science and Technology, Soochow University, Suzhou, China
[4]Harbin Institute of Technology, Shenzhen, China
{wuhongqiu,linfengliu}@sjtu.edu.cn,zhaohai@cs.sjtu.edu.cn,
minzhang@suda.edu.cn

## Abstract

Beyond the great cognitive powers showcased by language models, it is crucial to scrutinize whether their reasoning capabilities stem from strong generalization or merely exposure to relevant data. As opposed to constructing increasingly complex logic, this paper probes into the boolean logic, the root capability of a logical reasoner. We find that any pre-trained language models even including large language models only behave like a random selector in the face of multi-nested boolean logic, a task that humans can handle with ease. To empower language models with this fundamental capability, this paper proposes a new self-supervised learning method *Curriculum Logical Reasoning* (CLR), where we augment the training data with nested boolean logic chain step-by-step, and program the training from simpler logical patterns gradually to harder ones. This new training paradigm allows language models to effectively generalize to much harder and longer-hop logic, which can hardly be learned through naive training. Furthermore, we show that boolean logic is a great foundation for improving the subsequent general logical tasks[1].

## 1 Introduction

Artificial intelligence has made a giant leap from perception to cognition, with powerful pre-trained language models (PLMs) (Devlin et al., 2019; Liu et al., 2019; Lan et al., 2020; Clark et al., 2020; Raffel et al., 2020; Brown et al., 2020; He et al., 2021b), large language models (LLMs) (Chung et al., 2022; Chowdhery et al., 2022; OpenAI, 2023) demonstrating human-level comprehension and reasoning powers on a series of challenging tasks like commonsense reasoning (Zellers et al., 2019), open-domain question-answering (Mihaylov et al., 2018),

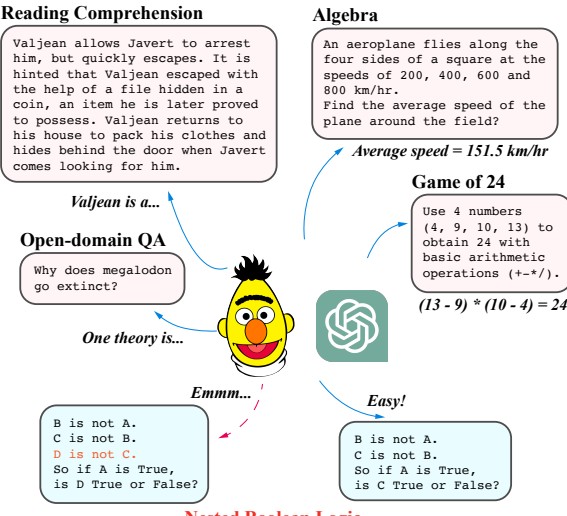

Figure 1: While language models are capable of handling a range of complex logical tasks, they do not perform well on more basic nested boolean logic.

arithmetical reasoning (Ling et al., 2017).

While this is charming, these over-parameterized language models are shown to be good at exploiting superficial statistical cues to achieve decent scores on end tasks (Zhou et al., 2021; Sanyal et al., 2022a; Wu et al., 2023b). Early on BERT, it is found that simply by adding a "not" to the claims, BERT would be fooled into a random selector (Niven and Kao, 2019). It is time to go back and scrutinize whether the state-of-the-art PLMs master solid logical capability, as truly powerful logical reasoners.

Rather than creating even more complex logic, this paper concentrates on the root level of logical reasoning - boolean logic, as in Figure 1. Any logic can be reduced to a combination of multiple boolean operations, including negation ¬, intersection ∧, and union ∨. In this paper, we introduce a new probing method to quantify the boolean logical reasoning of a language model, fine-grained to different levels of logical difficulty.

However, our results show that none of PLMs

---

*Corresponding author; This paper was partially supported by Joint Research Project of Yangtze River Delta Science and Technology Innovation Community (No. 2022CSJGG1400).

[1]https://github.com/gingasan/boolkill

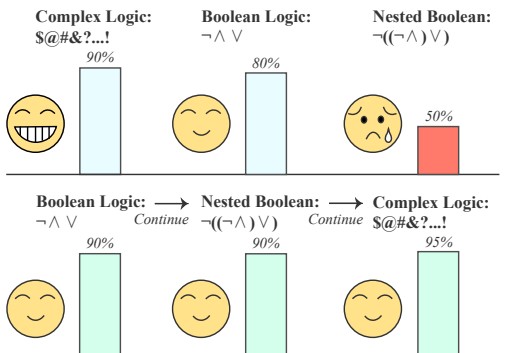

Figure 2: Overview of *Curriculum Logical Reasoning*.

| The earth is flat. | *Original sample* |
|---|---|
| $S_0$: The earth is flat. 
 Is $S_0$ true or false? | *Convert to context-question* |
| $S_0$: The earth is flat. 
 $S_1$: $S_0$ is a false statement. 
 $S_2$: $S_1$ is a false statement. 
 $S_3$: $S_2$ is a true statement. 
 Is $S_3$ true or false? | *Add nested boolean logic* 
 *NOT only* |
| $S_0$: The earth is flat. 
 $S_1$: $S_0$ is a false statement. 
 $S_2$: $S_1$ is a false statement. 
 $S_3$: Either $S_2$ or $S_1$ is a true statement. 
 / $S_3$: Both $S_2$ and $S_1$ are true statements. 
 Is $S_3$ true or false? | *Add nested boolean logic* 
 *NOT & AND & OR* |

Table 1: Method to augment arbitrary samples with nested boolean logic.

possess the necessary proficiency to tackle the multiple nesting of (multi-nested) simple boolean operations, even the state-of-the-art models like DeBERTa-V3 (He et al., 2021a) and ChatGPT (OpenAI, 2023). Faced with more than three nested boolean operations, they quickly degenerate into a random selector. even with the chain-of-thought prompt (Wei et al., 2022; Zhang et al., 2022b). Conversely, this task is very simple for humans, compared to other more general reasoning tasks. **This raises a shadow over their generalizability acquired from large amount of training.**

To empower the language models with such a fundamental capability in nested boolean logic, we propose a new self-supervised training paradigm, *Curriculum Logical Reasoning* (CLR), inspired by curriculum learning (Bengio et al., 2009). Concretely, we construct the nested boolean logic step-by-step from simple to hard on top of the original training samples in a self-supervised manner (Devlin et al., 2019). The model is encouraged to start with learning simple logical patterns and then move forward to hard ones gradually, rather than learning hard logic with a single leap. We find that recalling simpler logic while learning harder logic can result in a better outcome. Our experiments demonstrate that CLR significantly enhances the logical learning process. Excitingly, pre-learning boolean logic acts as a great foundation step to further enhance the subsequent logical end tasks, like ReClor and DREAM. Figure 2 illustrates CLR very lively.

## 2 Introducing Nested Boolean Logic

This section presents our method to introduce multi-nested boolean logic to existing data.

We first present the notations. Let $\mathbf{x}$ denote the input text, with its ground truth $y$, and $p_\theta$ denote the classifier (e.g. a language model) with parameters $\theta$. Given an arbitrary input sample $\mathbf{x}$, suppose that the model accurately predicts $p_\theta(\mathbf{x}) = y$. We now define an operation $\delta$ on $\mathbf{x}$, which can be regarded as a transformation on the text, denoted as $\delta \cdot \mathbf{x}$.

### 2.1 From Simple Boolean Logic to Nested Boolean Logic

We concentrate on the logical operation, which specifically manipulates the underlying logical chain by transformation on the text. We present a new form of logical operation that corresponds to only boolean operators, i.e. intersection $\wedge$, union $\vee$, and negation $\neg$. We might concentrate on the simplest negation first.

Suppose that the input statement $\mathbf{x}$ entails a fact $f$, which can be either a true fact or a false fact, represented by $y_0$. The logical process can be formulated as $\mathbf{x} \Rightarrow y_0$, where $\Rightarrow$ refers to "implies that" and $y_0 \in \{0, 1\}$ (0 for *True* and 1 for *False*).

We illustrate a toy example of our logical operation in Table 1. First, the model is required to discriminate whether the stated fact in $\mathbf{x}$ is true or false. It states a false fact "the earth is flat", so $y_0 = 1$ (*False*). Next, we transfer it to a context-question template and denote the context as $S_0$. It is still a binary classification and the answer for it is limited in *True* or *False*. This template can be applied to arbitrary tasks. For instance, a sentiment analysis sentence "cold movie" can be rewritten to a statement like "cold movie expresses a positive movie watching".

Our idea is to craft a series of statements after $S_0$. Each statement asserts the truth or falsity of the previous statement, which is uniformly chosen. We denote such a statement as *boolean statement*, and

ask the model to discriminate the final statement. For instance, $y_0 = 1$ and $S_1$ asserts $S_0$ is false, so $y_1$ should be negated, $y_1 = 0$. After deduction, we can obtain $y_3 = 1$.

Logically, the assertion of "true" results in no change of the current logic and the assertion of "false" results in a negation. $\delta$ can be nested for $k$ times without affecting the fact in $\mathbf{x}$:

$$\prod_{i=1}^{k} \delta_i \cdot \mathbf{x} \Rightarrow y_k \tag{1}$$

where $y_i$ denotes each intermediate answer after $i$ times of boolean statements and $y_k$ denotes the eventual answer. We denote Eq. 1 as *multi-nested boolean logic*.

Obtaining final $y_k$ is free of external annotation, as in self-supervised learning, by programming the following recursion:

$$y_i = \begin{cases} \neg \quad y_{i-1} & , \ \delta_i \ \text{asserts false} \\ \quad y_{i-1} & , \ \delta_i \ \text{asserts true} \end{cases} . \tag{2}$$

Such multi-nested boolean logic poses little challenge to humans. We hopefully assume that a strong language model can tackle that as well.

We generalize the negation operation to other boolean operations as in the bottom of Table 1. Concretely, we uniformly choose one statement from $S_1$ to $S_k$ and append it with either "and" or "or" chosen uniformly.

## 2.2 Quantify Boolean Logic

We probe the mastery in nested boolean logic of a language model by measuring its performance against our boolean statements. An ideal logical reasoner is supposed to make clear logical transitions between truth and falsity. We are particularly interested in this situation: **the model accurately discriminates the original fact, while falters in delivering the correct answer subsequent to $k$ boolean statements.** This can be formulated as:

$$p_\theta \left( \prod_{i=1}^{k} \delta_i \cdot \mathbf{x} \right) \neq y_k \tag{3}$$

where $p_\theta$ satisfies:

$$p_\theta (\mathbf{x}) = y_0. \tag{4}$$

Deep neural models are good at exploiting superficial features rather than delving into the entire semantics (Wu et al., 2023a; Sanyal et al., 2022a).

The consequence is that they can get the final result without correctly classifying the original fact. Eq. 3 and 4 exclude this potential threat and focus entirely on the model's capability in handling nested boolean logic. In other words, if the model reasons from a misclassified fact, its final result can be noisy, misleading the analysis.

Hence, we are interested in two metrics:

• *Clean accuracy (clean%)*: It refers to the general accuracy score.

• *Boolean accuracy (boolean%)*: It refers to the accuracy only calculated on those samples where the model accurately discriminates the original fact, as represented in Eq. 3 and 4. This can only be calculated on augmented data.

## 3 Benchmark

To benchmark the multi-nested boolean logic, we construct a new dataset in this paper and following experiments are based on this. As apart from other datasets, it is composed of a series of subsets, representing different levels of logical complexity. We will release this benchmark for future research.

### 3.1 Data Collection

We collect the raw data from SciTail (Khot et al., 2018), a scientific text entailment dataset with a premise and a hypothesis for each sample, which is labeled as *entail* or *not entail*. We join the premise and hypothesis together to make them a "fact", with the entailed pair labeled as *True* and not entailed one labeled as *False*. Some samples are shown in Appendix A. Eventually, we get 6,000 raw samples and randomly sample 1,000 of them as the test set with the rest as the training set.

On top of the raw data, we convert it to the context-question format and then impose boolean statements to generate the adversarial set, which means that the resultant samples are likely to fool the model (Zellers et al., 2018, 2019). Specifically, we uniformly choose a value $k$ from some range and insert $k$ boolean statements following the original sample. The range of $k$ bounds the minimal and maximal nesting of boolean logic on each sample, and larger value of $k$ suggests more nesting on the logic chain. For instance, the samples in Table 1 correspond to $k = 0$ and $k = 3$ (see Appendix A).

We denote this benchmark as *BoolKill*, in which each sample is a logic chain started with a potential fact and followed by a series of boolean statements. It is worth noting that BoolKill is a group of sets

|         | DeBERTa-base | DeBERTa-large | GPT2-1.5b |
|---------|--------------|---------------|-----------|
| $raw$   | 96.4         | 98.1          | 96.0      |
| $u_0$   | 96.7         | 97.8          | 96.8      |

Table 2: Performances on raw data and its templated $u_0$.

for different levels of logical difficulty, and each level has its own training and test set. We use the following notations to spot them:

• $raw$: the raw data in which each sample is a statement of a fact;

• $u_0$: the clean set in which each raw sample is only transferred to a context-question template, with semantics unchanged;

• $u_k$: the adversarial set constructed on top of $u_0$ in which each sample is suffixed by $k$ boolean statements;

• $u_{k_1 \sim k_2}$: the adversarial set in which each sample is suffixed by $k_1 \sim k_2$ boolean statements;

• $\tilde{u}_k / \tilde{u}_{k_1 \sim k_2}$: $u$ is negation-only, and we use $\tilde{u}$ to distinguish the adversarial set additionally containing AND and OR.

### 3.2 Data Bias

The first thing to verify is whether $u_0$ is semantically equivalent to $raw$. From Table 2, we find that each model achieves very close performances on $raw$ and $u_0$, suggesting that the context-question template does not induce bias to the original data.

The average sentence length will vary due to the boolean statements on raw data, which grows linearly from 36 to 88, from $u_1$ to $\tilde{u}_8$. The overall statistics of BoolKill are in Appendix A.

To minimize the bias between subsets, we keep the ratio of positive and negative samples to 1:1 in all subsets. Additionally, BoolKill is a semi-annotated dataset, comprising human-annotated facts and synthetic boolean statements. The latter introduces several high-frequency words like "true", "false", and "statement", which may induce large bias if these words do not occur in balance in data. For instance, the model may make the decision based on the relative number of "true" and "false" in the sentence. Hence, we also keep the occurrence of "true" and "false" the same for both positive and negative samples in all subsets.

### 3.3 Evaluation Results

We report the thorough results on each level of logical difficulty on BoolKill. We sequentially evaluate each model on $u_0$, $u_1$, $u_2$, ..., and $u_8$ ($\tilde{u}_8$), indicating the number of nested boolean operations.

We evaluate two state-of-the-art PLMs:

• DeBERTa-V3 (He et al., 2021a): one of the strongest BERT-style language models;

• ChatGPT (OpenAI, 2023): the strongest large language model, as a powerful zero-shot learner.

ChatGPT shows an impressive ability to follow human instructions and we directly evaluate it on the test sets[2]. For DeBERTa, we first fine-tune it on the $u_k$ training set and evaluate it on the $u_k$ test.

**NOT:** We curve the results in Figure 3. We find that each model exhibits a high performance on $u_1$, suggesting their proficiency in tackling single boolean logic. DeBERTa performs better than ChatGPT, probably due to task-specific fine-tuning. However, as the nesting increases, each model suffers from a notable decline regardless of size. For instance in (a), starting from $u_2$, in which the samples are suffixed by only two boolean statements, DeBERTa-base falls to 53.8% while DeBERTa-large falls to 65.4%. From $u_3$, strong as DeBERTa-large, it leans to a random selector, whose accuracy gets close to 50%. Similar situations can be seen on ChatGPT, while its degradation is more gentle. It suggests that even state-of-the-art models possess a critical limitation in the basic nested boolean logic, only able to handle up to three nested operations. This is far below humans' level.

**AND & OR:** From (b), it is counter-intuitive that DeBERTa performs better on sets additionally including AND and OR. We conjecture that the model utilizes the inherent bias that AND $\Rightarrow$ *False* and OR $\Rightarrow$ *True* in majority of cases. Such a shortcut is particularly useful when $k$ is small. Interestingly from (d), well-trained ChatGPT appears not to use this, and its performance drops even faster on $\tilde{u}$. Therefore, we focus on $\tilde{u}$ and $\tilde{u}$ with large $k$ in the following experiments.

Chain-of-Thought (CoT) (Wei et al., 2022; Zhang et al., 2022b; Yao et al., 2023) is proven to be an effective prompt method to amplify the reasoning ability of LLMs, with asking them to offer the procedure while performing the reasoning. From Figure 3 (c) and (d), we find that ChatGPT performs better with the assistance of CoT. However, we raise a criticism in the paper: does CoT promote logical reasoning? Indeed, our study show that CoT may bring new logical concern. We will

---

[2]We use the API from *openai*. The backbone model is *gpt-3.5-turbo*.

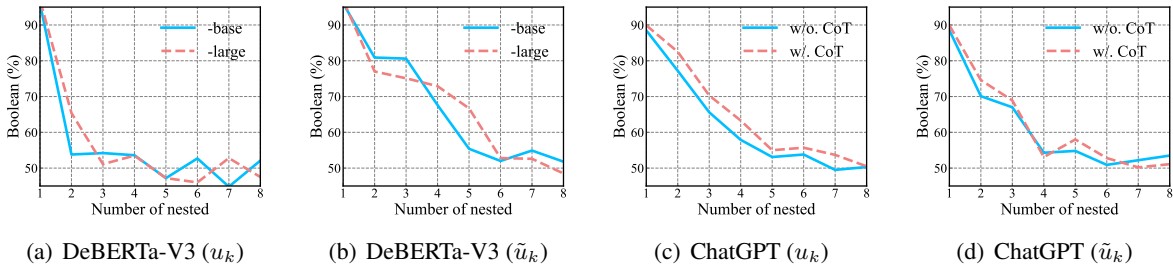

| (a) DeBERTa-V3 ($u_k$) | (b) DeBERTa-V3 ($\tilde{u}_k$) | (c) ChatGPT ($u_k$) | (d) ChatGPT ($\tilde{u}_k$) |

Figure 3: Boolean accuracy of different models with increasing numbers of nested boolean operations ($u_k/\tilde{u}_k$).

further discuss it in Sec. 6.1.

## 4 Empower Nested Boolean Logic

We present a new self-supervised learning manner.

### 4.1 Self-Supervised Learning

The straightforward method is to fine-tune the model on BoolKill. The concept behind is to sequentially introduce boolean statements on top of some corpus and let the model learn to tackle multi-nested boolean logic self-supervisedly.

However, we find language models struggle to fit the samples in BoolKill when the potential logic within the data is too hard, and still be a random selector. It indicates that naive training is not the best therapy to learn complex logical patterns.

### 4.2 Curriculum Logical Reasoning

Inspired by Curriculum Learning (Bengio et al., 2009), where the machine learning model is encouraged to learn the task starting with easier samples and ending with harder ones, we propose *Curriculum Logical Reasoning* (CLR) to enhance the process of learning logical reasoning.

There is a natural match between curriculum learning and logical philosophy, because the logic chain is a step-by-step progression from single to complex. CLR means that, rather than learning hard logic from scratch, the model starts with learning simpler logic, e.g. single boolean logic, and then moves forward to harder logic gradually, e.g. multinested boolean logic.

We show a concrete instance. We start to train the model on $u_{0\sim1}$, which solely includes single boolean operations. Next, we train such a model on $u_{0\sim2}$, which further includes two-nested boolean operations. This gradual progression continues until the model is trained on $u_{0\sim4}$. The above procedure can be denoted as $u_{0\sim1} \rightarrow u_{0\sim2} \rightarrow u_{0\sim3} \rightarrow u_{0\sim4}$. We find that reusing the easier

samples in the new turn of training benefits the eventual performance, which potentially reminds the model of what it learns previously. Our ultimate goal is that the model can gradually learn to tackle more complex logic that it has not seen before.

## 5 Empirical Results

As opposed to the prior section, where we evaluate the model on each level of logical difficulty, in this section, we evaluate each model on BoolKill $u_{1\sim4}$, $u_{5\sim8}$, and $\tilde{u}_{5\sim8}$ as an alternative. These sets cover the range from $k = 1$ to $k = 8$. $u_{1\sim4}$ is a simpler one and $u_{5\sim8}$ and $\tilde{u}_{5\sim8}$ appear to be highly challenging, since we previously show that state-of-the-art PLMs are almost powerless for the nested boolean logic beyond $u_3$.

We experiment on DeBERTa-V3-base and DeBERTa-V3-large. Each model is trained for 3,000 steps with a batch size of 16 and learning rate of 2e-5 / 1e-5 for the base / large one.

To verify CLR, we report two experiments. In the first experiment, we compare different training settings and evaluate the models on BoolKill. In the second, we leverage the boolean logic in BoolKill to benefit other general logical tasks.

### 5.1 Nested Boolean Logic

The results across various BoolKill sets are summarized in Table 3. We find that naively training the model on $u_{5\sim8}$ only produces random accuracy scores on all three test sets, even on two simpler ones $u_0$ and $u_{1\sim4}$. While on $u_{0\sim4}$ and $\tilde{u}_{0\sim4}$, we find that DeBERTa-V3-large can achieve better outcomes on simpler $u_{1\sim4}$ compared to DeBERTa-V3-base. It suggests that a larger model possibly has a greater learning ability to handle more nested boolean operations, but it is still very hard even for strong DeBERTa-V3-large, to learn very difficult logical patterns in $u_{5\sim8}$ within a single leap.

However, CLR brings significant performance

| | | $u_0$ clean% | $u_{1\sim4}$ | $u_{5\sim8}$ boolean% | $\tilde{u}_{5\sim8}$ |
|---|---|---|---|---|---|
| | | | | *DeBERTa-V3-base* | |
| Naive | $u_{5\sim8}$ | 50.2 | 46.6 | 49.6 | 51.2 |
| | $u_{0\sim4}$ | 96.0 | 71.5 | 53.6 | 53.4 |
| | $\tilde{u}_{0\sim4}$ | 94.6 | 72.2 | 52.9 | 56.0 |
| | $u_{0\sim1}$ | 96.2 | 64.6 | 49.0 | 50.5 |
| | $\to u_{0\sim2}$ | **96.4** | 89.6 ↑ | 57.6 ↑ | 56.7 ↑ |
| CLR | $\to u_{0\sim3}$ | 96.1 | 94.6 ↑ | 70.0 ↑ | 60.7 ↑ |
| | $\to u_{0\sim4}$ | 96.3 | 96.8 ↑ | **79.2** ↑ | 66.8 ↑ |
| | $\to \tilde{u}_{0\sim4}$ | 95.8 | **97.4** ↑ | 77.5 | **73.0** ↑ |
| | | | | *DeBERTa-V3-large* | |
| Naive | $u_{5\sim8}$ | 55.2 | 48.6 | 51.3 | 50.8 |
| | $u_{0\sim4}$ | 96.4 | 97.9 | 61.9 | 54.7 |
| | $\tilde{u}_{0\sim4}$ | 96.1 | 77.6 | 51.7 | 60.8 |
| | $u_{0\sim1}$ | 97.7 | 70.2 | 52.7 | 48.7 |
| | $\to u_{0\sim2}$ | **98.0** | 87.0 ↑ | 60.7 ↑ | 55.2 ↑ |
| CLR | $\to u_{0\sim3}$ | 97.7 | 98.5 ↑ | 71.1 ↑ | 59.3 ↑ |
| | $\to u_{0\sim4}$ | 97.6 | 99.4 ↑ | **84.3** ↑ | 68.2 ↑ |
| | $\to \tilde{u}_{0\sim4}$ | 97.3 | **99.5** ↑ | 81.9 | **82.3** ↑ |

Table 3: Results on BoolKill, comparing CLR with naive training. We use "→" to denote the curriculum setting we perform, where the model inherits the trained weights from the last level. We highlight the step-by-step performance gains CLR brings with "↑".

| | | ReClor | DREAM |
|---|---|---|---|
| | $sp$ | 58.2 | 79.9 |
| *DeBERTa* | $u_0 \to sp$ | 59.0 | 80.2 |
| *-V3-base* | $u_{0\sim1} \to sp$ | 61.6 ↑3.4 | 82.0 ↑2.1 |
| | $u_{0\sim1} \to u_{0\sim2} \to sp$ | **62.6** ↑4.4 | **82.8** ↑2.9 |
| *DeBERTa* | $sp$ | 71.4 | 90.4 |
| *-V3-large* | $u_{0\sim1} \to u_{0\sim2} \to sp$ | **74.8** ↑3.4 | **92.5** ↑2.1 |
| *LLaMA2* | $sp$ | 55.4 | 85.1 |
| *-7b (LoRA)* | $u_{0\sim1} \to u_{0\sim2} \to sp$ | **61.6** ↑6.2 | **86.9** ↑1.8 |
| | | $u_{0\sim1}$ | |
| *DeBERTa* | $u_{0\sim1}$ | 96.6 | 98.1 |
| *-V3-base* | $sp \to u_{0\sim1}$ | 95.3 ↓1.3 | 97.8 ↓0.3 |

Table 4: Results on general MRC tasks. "$sp$" refers to the task-specific training set and we evaluate the model on the corresponding test set.

(Yu et al., 2020), a reasoning-required MRC collected from graduate admission exams; • DREAM (Sun et al., 2019), a dialogue-based MRC. Concretely, we first train DeBERTa-V3 on BoolKill as an initialization and then fine-tune it on the task-specific data of ReClor and DREAM.

The results are shown in Table 4. We find that learning boolean logic acts as a nice initialization for the subsequent reasoning tasks on both ReClor and DREAM. For instance, initializing with $u_{0\sim1}$ improves DeBERTa-V3-base by 3.4% compared to naive fine-tuning on ReClor, and $u_{0\sim1} \to u_{0\sim2}$ further improves by 4.4%. It is worth noting that $u_0$ alone does not provide any useful signals (59.0% on ReClor and 80.2% on DREAM), suggesting that it is the boolean logic that we add into the data that enhances the eventual logical performance.

As a contrast, we first train the model on task-specific data and then fine-tune it on boolean logic. We find that more complex logic in ReClor or DREAM does not enable the model to perform any better on $u_{0\sim1}$ or even harms it, confirming our initial idea, that the model may ignore the basic logic during training, even if it appears to handle more complex problems sometimes.

It is the generic form of CLR to pre-learn boolean logic and then learn complex logic.

### 5.3 Ablation Study

The ablation study is made under negation-only sets. We first discuss the composition of levels to make up the curriculum to perform CLR. We remove some levels from the full curriculum setting $u_{0\sim1} \to u_{0\sim2} \to u_{0\sim3} \to u_{0\sim4}$. Additionally, we include another strong baseline by merging all the

boosts on every model and every test set. Its advantages are especially significant on harder $u_{5\sim8}$ and $\tilde{u}_{5\sim8}$. For instance on DeBERTa-V3-large, it achieves an impressive boolean accuracy of 84.3% on $u_{5\sim8}$ and 82.3% on $\tilde{u}_{5\sim8}$, uplifting naive training by about 30%, also keeping a high clean accuracy of 97.6% and 97.3% on $u_0$. It is worth noting that the model has not ever seen the hard samples in $u_{5\sim8}$ and $\tilde{u}_{5\sim8}$ during training, and CLR effectively generalizes the model to unseen logical patterns. Additionally, all models consistently maintain a strong accuracy on $u_0$ throughout the process of CLR, suggesting that they learn to discriminate the original facts and tackle boolean logic simultaneously. As a contrast, naive self-supervised training leads to inferior $u_0$ results.

Moreover, we find that each level of curriculum brings a considerable improvement to the model. For instance, the performance of DeBERTa-V3-base has outperformed all naive baselines when it just completes the second level of training on $u_{0\sim2}$.

### 5.2 Boolean Benefits Complex Logic

Boolean logic acts as the atomic component of logic. Our intuition is that it can solidify more general end tasks that require complex logical reasoning. We conduct validation on two machine reading comprehension (MRC) datasets: • ReClor

| | | $u_0$ clean% | $u_{1\sim4}$ boolean% | $u_{5\sim8}$ |
|---|---|---|---|---|
| NAI | $u_{0\sim1},\cdots,u_{0\sim4}$ | 95.5 | 95.8 | 66.5 |
| CLR | $u_{0\sim1}\to\cdots\to u_{0\sim4}$ | **96.3** | **96.8** | **79.2** |
| NAI | $u_{0\sim1},u_{0\sim3}$ | 95.4 | 86.6 | 55.7 |
| CLR | $u_{0\sim1}\to u_{0\sim3}$ | 95.8 | 92.7 | 66.4 |
| NAI | $u_{0\sim2},u_{0\sim4}$ | 95.8 | 60.1 | 51.1 |
| CLR | $u_{0\sim2}\to u_{0\sim4}$ | 90.8 | 82.9 | 55.8 |
| NAI | $u_{0\sim1},u_2,u_3,u_4$ | 95.6 | 95.6 | 65.6 |
| | $u_{0\sim1}$ | **96.2** | 64.6 | 49.0 |
| CLR | $\to u_2$ | 95.9 | 89.9 | 57.7 |
| | $\to u_3$ | 95.3 | 94.3 | 64.8 |
| | $\to u_4$ | 95.6 | **96.5** | 72.2 |

Table 5: Ablation study on DeBERTa-V3-base. We omit the notations of $u_{0\sim2}$ and $u_{0\sim3}$ in "$\cdots$".

training sets together, e.g. $u_{0\sim1},u_{0\sim2},u_{0\sim3},u_{0\sim4}$ and performing naive training. The difference is that CLR strategically samples the training data from easy ones to hard ones rather than uniformly. The results are summarized in Table 5. We find that any leap from the full curriculum can result in a notable performance drop, highlighting the importance of a complete and gradual progression of logical learning. Interestingly, we also find that learning from simpler $u_{0\sim1}\to u_{0\sim3}$ achieves a better outcome compared to harder $u_{0\sim2}\to u_{0\sim4}$.

Next, we discuss the composition of samples for each level. We remove the simpler samples that belong to the prior level ($u_{0\sim1}\to u_2\to u_3\to u_4$) and see whether the model would forget what it has learned before as a result. From Table 5, we find that the removal process gives comparable results on $u_0$ and $u_{1\sim4}$, However, when it comes to harder $u_{5\sim8}$, it leads to a performance drop of 6%. These findings underscore the importance of reusing simpler samples when stepping forward to the new level, especially when evaluating on harder or even unseen data like $u_{5\sim8}$.

## 5.4 Fine-tuning Large Language Models

We also evaluate our method on LLMs. However, fine-tuning LLMs requires a huge amount of resources. As a compromise, recent studies propose several efficient fine-tuning methods that only update a small ratio of parameters within LLMs. We experiment on three models, GPT2-1.5b (Brown et al., 2020), OPT-7b (Zhang et al., 2022a), and LLaMA2-7b (Touvron et al., 2023). They both belong to the decoder-only architecture as ChatGPT. We fine-tune GPT2-1.5b with full parameters and fine-tune the 7b models with the low rank adaption

| | | $u_0$ clean% | $u_{1\sim4}$ boolean% | $u_{5\sim8}$ |
|---|---|---|---|---|
| | | | *GPT2-1.5b* | |
| NAI | $u_{0\sim1},\cdots,u_{0\sim4}$ | 93.8 | 99.2 | 65.6 |
| | $u_{0\sim1}$ | **95.6** | 74.0 | 52.8 |
| CLR | $\to u_{0\sim2}$ | 94.4 | 84.6 | 55.8 |
| | $\to u_{0\sim3}$ | 94.1 | 98.6 | 71.2 |
| | $\to u_{0\sim4}$ | 94.3 | **99.9** $_{\uparrow0.7}$ | **79.4** $_{\uparrow13.8}$ |
| | | | *OPT-7b (LoRA)* | |
| NAI | $u_{0\sim1},\cdots,u_{0\sim4}$ | 94.3 | 98.0 | 63.8 |
| | $u_{0\sim1}$ | 93.3 | 68.7 | 54.2 |
| CLR | $\to u_{0\sim2}$ | 94.3 | 78.2 | 53.0 |
| | $\to u_{0\sim3}$ | 94.7 | 97.9 | 64.5 |
| | $\to u_{0\sim4}$ | **95.5** | **98.8** $_{\uparrow0.8}$ | **69.4** $_{\uparrow5.6}$ |
| | | | *LLaMA2-7b (LoRA)* | |
| NAI | $u_{0\sim1},\cdots,u_{0\sim4}$ | 97.3 | 99.4 | 67.9 |
| | $u_{0\sim1}$ | 96.3 | 64.3 | 48.3 |
| CLR | $\to u_{0\sim2}$ | 97.6 | 86.2 | 51.8 |
| | $\to u_{0\sim3}$ | **97.7** | 98.6 | 67.8 |
| | $\to u_{0\sim4}$ | 97.6 | **99.9** $_{\uparrow0.5}$ | **75.9** $_{\uparrow8.0}$ |

Table 6: Results of LLMs, including the efficient fine-tuning method (LoRA).

method (LoRA) (Hu et al., 2022).

From Table 6, we find that CLR works very well on GPT2-1.5b, achieving a boolean accuracy of 79.4% on $u_{5\sim8}$, outperforming naive training by a notable margin of 13.8%. However, larger-scaled OPT-7b does not yield better results as expected. Specifically, it achieves comparable results on simpler $u_{1\sim4}$, while greatly lags behind much smaller GPT2-1.5b on harder $u_{5\sim8}$. We conjecture that parameter efficient fine-tuning might compromise the acquisition of complex reasoning capability, e.g. multi-nested boolean logic, leading to a non-negligible performance drop.

## 6 Further Discussion

### 6.1 Chain-of-Thought

We discuss CoT in more detail, and this part is particularly geared to the current LLMs. It has been shown that when being asked to give the procedure, the model can perform the reasoning more precisely. In the prior section, we show that CoT can assist ChatGPT in achieving better performance on BoolKill. We notice that the intermediate thinking procedure exposed by CoT is equally important. A ideal reasoner can not only make the final answer but also reasonable intermediate results.

However, we find that ChatGPT leans to fall into inconsistent deduction when giving the intermediate results, as we illustrate in Table 7.

For $[a]$, we can first obtain that $S_3$ is false from

the previous statements since $S_0$ is true. Hence, the deduction of the model up to this step is correct. In the next step, however, the model draws an incorrect conclusion based on the fact that $S_3$ is false, that is $S_4$ is true. Indeed, $S_4$ should be false since it doesn't match $S_3$, incurring a wrong final answer. Similar cases can be found in [b]. These cases indicate that when making longer reasoning, ChatGPT can fall into mistake in some logical step, even though each step is very easy when cutting individually.

---

[a] *Query*

---

S0: A crust is a portion of a world.
S1: S0 is a false statement.
S2: S1 is a false statement.
S3: S2 is a false statement.
S4: S3 is a true statement.
Is S4 true or false? Let's think step by step.

---

[a] *Answer*

---

...
S3: S2 is a false statement.
It contradicts the prior assessment S2 is true, so S3 is false.
S4: S3 is a true statement.
It contradicts the prior assessment S3 is false, so S4 is true.

---

[b] *Query*

---

S0: Humans sometimes eat clouds.
S1: S0 is a true statement.
S2: S1 is a false statement.
S3: S2 is a true statement.
S4: S3 is a true statement.
S5: S4 is a false statement.
Is S5 true or false? Give the procedure.

---

[b] *Answer*

---

...
S1: S0 is a true statement. (True assuming S0 is true)
S2: S1 is a false statement. (False because S1 is true)
S3: S2 is a true statement. (True because S2 is false)
S4: S3 is a true statement. (False because S3 is true)
...
Therefore, S5 is true based on the evaluation.

---

Table 7: ChatGPT case study. $S_4$ in [a] should be false.

## 6.2 True or False

We take a further look at true-or-false questions, a specific and common question type in MRC and logical end tasks. Specifically, we filter out the samples with questions that contain keywords "true" or "false". In ReClor, there are 173 such samples out of the 500 in its development set. The evaluation results on true-or-false questions are shown in Table 8. We find that both DeBERTa models struggle with seemingly simple true-or-false questions, showing lower accuracy compared to the overall performance. However, the models pre-learned

| | | True/False | All |
|---|---|---|---|
| *DeBERTa-V3-base* | *sp* | 51.4 | 58.2 |
| *DeBERTa-V3-base* | *boolean → sp* | **57.8** | 62.6 |
| *DeBERTa-V3-large* | *sp* | 67.1 | 71.4 |
| *DeBERTa-V3-large* | *boolean → sp* | **73.4** | 74.8 |

Table 8: Results on true-or-false questions in ReClor.

with nested boolean logic showcase a significant improvement, achieving 6.4% and 6.3% points of gain respectively.

## 7 Related Work

The study of boolean operations is the fundamental requirement for a series of challenging tasks, e.g. arithmetical reasoning (Ling et al., 2017), commonsense reasoning (Zellers et al., 2019), reading comprehension (Yang et al., 2018), dialogue comprehension (Sun et al., 2019). We concentrate on the multi-nested boolean logic by augmenting the text with boolean statements. Previous studies analyze more general logical reasoning, e.g. RICA (Zhou et al., 2021), RobustLR (Sanyal et al., 2022a), FaiRR (Sanyal et al., 2022b), by logical paraphrase or contrast sets.

Self-supervised learning methods typically generate learnable inputs on top of unlabeled corpora, e.g. by masking (Devlin et al., 2019), insertion (Wu et al., 2022), sentence reordering (Lan et al., 2020), contrastive learning (Gao et al., 2021), while our method is by imposing a series of sentences to the suffix, actually generating learnable logic. We introduce curriculum learning (Bengio et al., 2009), which allows the model to learn step by step to further facilitate self-supervised learning. Curriculum learning is under-discussed in context of language processing (Xu et al., 2020; Lee et al., 2022).

While deep neural networks are capable of handling very complex tasks, in reality they lean to exploit spurious cues (Goodfellow et al., 2015; Madry et al., 2018; Wu et al., 2023a), and can be powerless to very simple perturbations as a consequence. Our work discloses that language models are poorly skilled at basic boolean logic. In parallel, studies show that language models can be easily fooled by some naive patterns within the text, e.g. lexical overlap (McCoy et al., 2019; Wu et al., 2023c), entity boundary (Yang et al., 2023), word order (Zhang et al., 2019).

We also release a challenging benchmark to evaluate boolean logical reasoning. There are a series of work focusing on constructing challenging logic,

e.g. ReClor (Yu et al., 2020), HotpotQA (Yang et al., 2018), ANLI (Nie et al., 2020).

## 8 Conclusion

This paper provides a quantified analysis on the multi-nested boolean logic. We flag the deficiency in the state-of-the-art language models in terms of such basic capability, which will inevitably cause pitfalls in dealing with more complex reasoning tasks. For this, we propose *Curriculum Logical Reasoning*, a new self-supervised learning method to empower language models with foundational logical capability. We also show that our idea can act as a cornerstone learning method for general logical reasoning.

## Limitations

We cannot exhaust all the arrangements of curriculum to perform CLR, which could potentially achieve even better performances. We have discussed the potential risk of chain-of-though as secondary contribution of our work, which will be interesting to study in the future. Our method to introduce nested boolean logic is general, while our experiments are based on one source. Another option is to collect data from more general corpus or specific domains of interest, which is promising. Eventually, we do not have enough resources to run large language models above 7b.

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

# A    BoolKill

| | Task | PN ratio | Size | Length | Vocab |
|---|---|---|---|---|---|
| **BoolKill** | binary | 1:1 | 6000 | 36~88 | 14315 |

Table 9: Statistics of BoolKill.

| | |
|---|---|
| SciTail | *[Premise]* The planet Mercury is the closest of the planets to the Sun. 
 *[Hypothesis]* Mercury is closest to the sun. 
 *[Label]* Entail |
| Context-question (k=0) | S0: The planet Mercury is the closest of the planets to the Sun. 
 So, Mercury is closest to the sun. 
 Is S0 true or false? 
 *[Label]* True |
| BoolKill (k=1) | S0: The planet Mercury is the closest of the planets to the Sun. 
 So, Mercury is closest to the sun. 
 S1 is a false statement. 
 Is S1 true or false? 
 *[Label]* False |
| BoolKill (k=3) | S0: The planet Mercury is the closest of the planets to the Sun. 
 So, Mercury is closest to the sun. 
 S1 is a false statement. 
 S2 is a true statement. 
 S3 is a false statement. 
 Is S3 true or false? 
 *[Label]* True |

Table 10: Illustration of some samples from BoolKill.