# OpenReview forum: "Empower Nested Boolean Logic via Self-Supervised Curriculum Learning"
_EMNLP/2023/Conference — EMNLP 2023 Main_

### Official Review · Reviewer_sk6i · 2023-08-04

**Soundness:** 3

**Excitement:**

4: Strong: This paper deepens the understanding of some phenomenon or lowers the barriers to an existing research direction.

**Missing References:**

N.A.

**Paper Topic And Main Contributions:**

**Paper Topic:**
The paper focuses on probing the logical reasoning capabilities of pre-trained language models, particularly their understanding and handling of multi-nested boolean logic. The study investigates the apparent gap between language models and human capabilities in reasoning and proposes a method to enhance the models' logical proficiency.

**Main Contributions:**
1. **Analysis of Existing Models**: The paper discovers that existing pre-trained language models, even large ones, behave like random selectors when faced with multi-nested boolean logic, which humans can handle with ease.

2. **Introduction of Curriculum Logical Reasoning (CLR)**: A new self-supervised learning method called Curriculum Logical Reasoning (CLR) is proposed. CLR augments training data with multi-nested boolean logical chains step-by-step, programming the training with simpler logical patterns before gradually moving to harder ones. This enables the model to learn much more complex and longer logic chains, which would be challenging through naive training.

3. **Enhanced Performance through CLR**: Experiments demonstrate that CLR significantly enhances the logical learning process in language models, allowing them to handle more complex reasoning tasks.

5. **Additional Findings**: The paper also presents several additional insights:
   - Learning harder logic while recalling simpler logic leads to better outcomes.
   - A chain-of-thought approach exposes basic logical flaws but can improve final answers.
   - Progressively offering samples from simple to hard stabilizes the performance of in-context learning.

Overall, the paper offers a comprehensive analysis of the limitations in current language models' reasoning capabilities and provides a new approach to improving them through a graduated and systematic learning paradigm. I will consider to raise scores if the authors address my concerns.

**Questions For The Authors:**

1. Why do the authors choose OPT-7B instead of LLaMA-7B for the experiments?
2. Could you explain a bit more about the following conclusion in Section 5.3: "These findings underscore the importance of reusing simpler samples when stepping forward to the new level, especially when evaluating on harder or even unseen data like u5∼8."?
3. Could you verify whether the conclusion "Learning boolean logic serves as a foundational step that further improves subsequent tasks like ReClor and DREAM." holds for large language models?

**Reasons To Accept:**

1. The author built the BoolKill benchmark for the evaluation of Nested Boolean Logic and conducted comprehensive experiments to show the improvement across different sizes of language models.
2. The author found that Learning boolean logic serves as a foundational step that further improves subsequent tasks like ReClor and DREAM based on the experiments on DeBERTa.

**Reasons To Reject:**

1. The authors of this paper made an earnest attempt to encompass numerous aspects simultaneously. However, the overall cohesion of the paper lacks a distinct logical progression. I would strongly advise the authors to reconsider the organization of their presentation.
2. In need of improvement is the section dedicated to related work (Section 7), which appears somewhat disorganized. I recommend that the authors consider revising Section 7 to enhance its logical coherence.
3. In Section 6.2, the authors' evaluation was based on only 10 samples. This limited sample size may have resulted in biased findings due to the small scale of the experiment.
4. I am concerned about 5.4 Fine-tuning Large Language Models regarding the model size. Is GPT2-1.5B qualified for LLMs?

**Reproducibility:**

4: Could mostly reproduce the results, but there may be some variation because of sample variance or minor variations in their interpretation of the protocol or method.

**Reviewer Confidence:**

4: Quite sure. I tried to check the important points carefully. It's unlikely, though conceivable, that I missed something that should affect my ratings.

**Typos Grammar Style And Presentation Improvements:**

Certain sentences appear unnatural, such as the following example: "When being asked to give the procedure, the model can perform more precisely the tasks that require logical reasoning."

---

> ### Author Rebuttal · Authors · 2023-08-24
>
> Thank you for your review.
>
> **Q1 & 3**: Results on LLaMA-7b
>
> Thank you for your suggestion. We have made the experiments on the latest LLaMA 2 model. Please see the results below, which highly support the conclusions in our paper.
>
> | | | $u_0$ | $u_{1\sim4}$ | $u_{5\sim8}$ |
> | :---- | ----: | :----: | :----: | :---- |
> | OPT-7b (LoRA) | |
> | Naïve | $u_{0\sim1}$, $u_{0\sim2}$, $u_{0\sim3}$, $u_{0\sim4}$ | 94.3 | 98.0 | 63.8 |
> | CLR | $u_{0\sim1}$ | 93.3 | 68.7 | 54.2 |
> | | -> $u_{0\sim2}$ | 94.3 | 78.2 | 53.0 |
> | | -> $u_{0\sim3}$ | 94.7 | 97.9 | 64.5 |
> | | -> $u_{0\sim4}$ | **95.5** | **98.8** | **69.4**  $\uparrow5.6$ |
> | LLaMA2-7b (LoRA) | |
> | Naïve |  $u_{0\sim1}$, $u_{0\sim2}$, $u_{0\sim3}$, $u_{0\sim4}$ | 97.3 | 99.4 | 67.9 |
> | CLR | $u_{0\sim1}$ | 96.3 | 64.3 | 48.3 |
> | | -> $u_{0\sim2}$ | 97.6 | 86.2 | 51.8 |
> | | -> $u_{0\sim3}$ | **97.7** | 98.9 | 67.8 |
> | | -> $u_{0\sim4}$ | 97.6 | **99.9** | **74.9** $\uparrow7.0$ |
>
> Furthermore, pre-learning nested boolean logic benefits both subsequent DREAM and ReClor on LLaMA2-7b.
>
> | | | ReClor | DREAM |
> | :---- | :----: | :----: | :----: |
> | DeBERTa-large | sp | 71.4 | 90.4 |
> | DeBERTa-large | boolean + sp | 74.8 | 92.5 |
> | LLaMA2-7b (LoRA) | sp | 55.4 | 85.1 |
> | LLaMA2-7b (LoRA) | boolean + sp | **61.6** | **86.9** |
>
> We will add these results in our new revision and hope these supplementary results can further enhance the soundness of our paper. Thank you.
>
> **Q2:** Explain more on "These findings underscore the importance of reusing simpler samples…"
>
> We find that it is important to remind the model of what it previously learned when training on new samples. The improvement is especially significant when evaluating the model on harder logics. For instance, when the model is trained on $u_3$ (3-nested boolean logic), we incorporate $u_1$ and $u_2$ samples together into the training. The resultant model generalizes better to harder and unseen $u_{5\sim8}$. **Briefly, this strategy mitigates the forgetfulness in model training when performing CLR.**
>
>
> **R1:** Encompassing numerous aspects simultaneously affects the cohesion of the paper
>
> **We respectfully disagree with your opinion that our paper is not well-organized due to numerous aspects included, while our paper is recognized well-structured and easy to follow by multiple reviewers (e.g. ZqCq and hbvC).** Of course, we will keep trying our best to make our paper clear to all readers.
>
> Please allow us to re-clarify the contributions of our paper. **Our contributions are multi-aspect.**
>
> 1.	We introduce a new benchmark and offer a comprehensive analysis on boolean logical capability of PLMs/LLMs.
>
> 2.	We propose a new self-supervised learning method to enhance logical learning.
>
> 3.	Our method further enhances the logical capability on general logical tasks.
>
> 4.	Additional insights are provided, e.g. on CoT.
>
> **All these aspects above are very closely related to each other,** especially every posterior aspect can be right built on the basis of prior one, **which are tied to be a comprehensive entirety by nature,** serving for the core theme of our paper, logical reasoning.
>
> **We are glad to see your acknowledgement on the comprehensiveness of our work. If such acknowledgement makes sense, then there is no way such comprehensiveness making our work not sound.**
>
> Surely our work cannot be perfect, thanks for your comments to let our work better.
>
> **R2:** Related work
>
> **Due to our multi-aspect contributions, our related work should also involve multiple aspects of work, which is what we have done in the paper**, e.g. logical reasoning, self-supervised learning, curriculum learning, some benchmarks.
>
> **R3:** Small scale experiment in additional exploration of in-context learning on ChatGPT
>
> **To reduce the variance of the results, we perform three rounds of the same experiment on ChatGPT.** We conjecture that CLR facilitates in-context learning by empowering the model with more structured and diverse samples, e.g. easy to difficult, compared to all difficult.
>
> However, this is our additional exploration of CLR, **which is not the main contribution of our work.** While we realize that this part of the study is not complete, **our primary intention is to shed light to future research,** e.g. the better way to organize the samples for in-context learning.
>
> **We have already discussed this point in our Limitations section.** Thank you for your question.
>
> **R4:** Is GPT2-1.5b a large language model
>
> Typically, LLMs $\ge$ 7b, thus we experiment on OPT-7b in the paper.  **Our motivation to include GPT2-1.5b in our experiment is to compare the difference between LoRA and full-parameter fine-tuning,** since we do not have enough resources to conduct full-parameter fine-tuning on models above 7b.
>
> **Surely we have reported the LLaMA2-7b (LoRA) results as a supplementary for our experiment.**

---

### Official Review · Reviewer_hbvC · 2023-08-05

**Soundness:** 5

**Excitement:**

4: Strong: This paper deepens the understanding of some phenomenon or lowers the barriers to an existing research direction.

**Paper Topic And Main Contributions:**

This paper has two interesting sections, where it first evaluates the ability of language models to do boolean reasoning (not, and, or) via a newly created benchmark. Then, they propose a Curriculum-based training mechanism based on their proposed benchmark, which can enhance the ability of the same evaluated models on the reasoning tasks. They further showcase that pretraining on their dataset with their proposed mechanism can even enhance the ability of LMs on harder reasoning tasks.

**Questions For The Authors:**

- I was wondering whether you have analyzed the complex tasks as to how much of the reasoning capabilities from Boolkill are required in their question types and whether those are actually the questions that are getting answered correctly after using the pretrained models. (If the model struggled with negation before, did the pertaining help with that?!)

**Reasons To Accept:**

- The paper is well-written and easy to follow despite the included technical terms, the entangled process, and multi-aspect contributions.
- The results showcased in the paper strongly suggest their approach's effectiveness.

**Reasons To Reject:**

 - I do not find any reason to reject!

**Reproducibility:**

5: Could easily reproduce the results.

**Reviewer Confidence:**

4: Quite sure. I tried to check the important points carefully. It's unlikely, though conceivable, that I missed something that should affect my ratings.

**Typos Grammar Style And Presentation Improvements:**

- Line 231: I think it should be k boolean statements rather than `x`.

---

> ### Author Rebuttal · Authors · 2023-08-24
>
> Thank you for your insightful review and appreciation of our work.
>
> **Q1:** Whether we have analyzed the question type
>
> Thank you for your great question!
> We take a closer look at ReClor. We filter out the samples with the questions that contain the keywords "true" or "false".
> There are 173 such samples out of the 500 in its dev set.
>
> The results are below: ("sp" refers to task-specific training data)
>
> | | | True/False | All |
> | :---- | :----: | :----: | :----: |
> | DeBERTa-base | sp | 51.4 | 58.2 |
> | DeBERTa-base | boolean + sp | **57.8** | 62.6 |
> | DeBERTa-large | sp | 67.1 | 71.4 |
> | DeBERTa-large | boolean + sp | **73.4** | 74.8 |
>
> **It suggests that pre-learning nested boolean logic greatly enhances the performance of the model on "True/False"-type questions.**
>
> We promise to add this part of the experiment in our new revision. Thank you for your comments!
>
>
> Line 231 typo
>
> Thank you for your correction.

---

### Official Review · Reviewer_ZqCq · 2023-08-05

**Soundness:** 3

**Excitement:**

4: Strong: This paper deepens the understanding of some phenomenon or lowers the barriers to an existing research direction.

**Paper Topic And Main Contributions:**

This paper investigates whether pre-trained language models possess true logical reasoning capabilities and finds that they struggle with multi-nested boolean logic tasks. To address this issue, the authors propose a new self-supervised learning method called Clr, which gradually increases the complexity of logical patterns to enable language models to effectively generalize to much harder and longer-hop logic. The paper provides valuable insights into the limitations of pre-trained language models and proposes a new training method to empower language models with fundamental logical capabilities.

**Reasons To Accept:**

The paper is well-structured and provides a clear and thorough investigation into the logical reasoning capabilities of pre-trained language models. The proposed Clr method is a promising approach to enhance the logical reasoning abilities of language models, and the experimental results provide strong evidence to support this claim. Overall, the paper offers valuable insights into the limitations of pre-trained language models and proposes a new training method to address this issue. Future research can build on this work to further explore the potential of this approach in enhancing the logical reasoning capabilities of language models.

**Reasons To Reject:**

The proposed training process is complex, and it is worth considering how to integrate it seamlessly with the general pre-training process without compromising the model's ability to learn general capabilities. The paper provides valuable insights into enhancing the logical reasoning capabilities of language models, but the potential impact on the model's ability to learn general capabilities should be further explored. Future research can investigate how to balance the proposed training method with the general pre-training process to ensure that the model can learn both fundamental logical capabilities and general capabilities. Overall, the paper offers a promising approach to enhance the logical reasoning abilities of language models, but further research is needed to address the potential trade-offs between specialized and general capabilities.

**Reproducibility:**

4: Could mostly reproduce the results, but there may be some variation because of sample variance or minor variations in their interpretation of the protocol or method.

**Reviewer Confidence:**

2: Willing to defend my evaluation, but it is fairly likely that I missed some details, didn't understand some central points, or can't be sure about the novelty of the work.

---

> ### Author Rebuttal · Authors · 2023-08-24
>
> Thank you for your valuable review.
>
> **R1:** The proposed training process is complex
>
> Please allow us to clarify that our proposed **Curriculum Logical Reasoning (CLR) is a simple and easy-to-use training method.**
> Following our method, the nested boolean logic data can be obtained on top of almost any types of sentences (e.g. NLI, sentiment analysis), without additional annotation. Then, we train the model from easy level to hard level. The difficulty of all samples can be easily distinguished by the number of nesting.
>
> We will also share our data to further facilitate this process.
>
> **R2:** Potential trade-offs between boolean pre-training and general pre-training
>
> First, we agree that it would be interesting to study the combination of logical pre-training and general pre-training in future study. Thank you for your comments!
>
> However, we will answer your question from two perspectives.
>
> 1. **CLR is a continual training process after general pre-training like MLM**. Its purpose is to enhance the logical capability of LMs (e.g. performing better on logical end tasks). **Respectfully, the trade-off between the subsequent logical capability and general capability is not a critical issue,** to the best of our knowledge.
>
> Indeed, we acknowledge that any specialized training may compromise the general capability of an LM, **while this gap is not within the scope of our paper.**
>
> 2. In addition, CLR is better suited for the transfer learning in a multi-task setting as in T5. We assume that the ability to handle multiple tasks is also a general capability. In this case, logical reasoning is a crucial sub-task or capability. **Considering that our method greatly benefits tasks like ReClor and DREAM, it is highly promising that it will not compromise other tasks to a large extent.**
>
> Again, thank you for your question.

---

### Meta-Review · Area_Chair_AafW · 2023-09-15

**Recommendation:** 5

**Metareview:**

The paper explores the logical reasoning capabilities (or lack of thereof) of pre-trained language models, specifically in their comprehension and handling of multi-nested boolean logic. It addresses the disparity between language models and human reasoning skills and introduces a method to improve the models' logical proficiency. The study introduces a new benchmark and a curriculum-learning approach based on it. Reviewers found the work well-structured, clear and thorough, and having strong and comprehensive experimental results. Reviewers raised some minor issues concerning how catastrophic forgetting of the vanilla language modelling task could impact the performance of resulting models, and also the size of LLMs used. This issues were addressed during the rebuttals.

---

### Decision · Program_Chairs · 2023-10-07

**Decision:**

Accept-Main

**Comment:**

The paper explores the logical reasoning capabilities (or lack of thereof) of pre-trained language models, specifically in their comprehension and handling of multi-nested boolean logic. It addresses the disparity between language models and human reasoning skills and introduces a method to improve the models' logical proficiency. The study introduces a new benchmark and a curriculum-learning approach based on it. Reviewers found the work well-structured, clear and thorough, and having strong and comprehensive experimental results. Reviewers raised some minor issues concerning how catastrophic forgetting of the vanilla language modelling task could impact the performance of resulting models, and also the size of LLMs used. This issues were addressed during the rebuttals.